# Caterpillar: A Pure-MLP Architecture with Shifted-Pillars-Concatenation

### Jin Sun
School of Computer Science and
Engineering, University of Electronic
Science and Technology of China
Chengdu, China
sunjin@uestc.edu.cn

### Xiaoshuang Shi*
School of Computer Science and
Engineering, University of Electronic
Science and Technology of China
Chengdu, China
xsshi2013@gmail.com

### Zhiyuan Wang
School of Computer Science and
Engineering, University of Electronic
Science and Technology of China
Chengdu, China
yhzywang@gmail.com

### Kaidi Xu
Department of Computer Science,
Drexel University
Chengdu, United States
xu.kaid@husky.neu.edu

### Heng Tao Shen
School of Computer Science and
Engineering, University of Electronic
Science and Technology of China
Chengdu, China
Tongji University
Shanghai, China
shenhengtao@hotmail.com

### Xiaofeng Zhu
School of Computer Science and
Engineering, University of Electronic
Science and Technology of China
Chengdu, China
seanzhuxf@gmail.com

## Abstract

Modeling in Computer Vision has evolved to MLPs. Vision MLPs naturally lack local modeling capability, to which the simplest treatment is combined with convolutional layers. Convolution, famous for its sliding window scheme, also suffers from this scheme of redundancy and lower parallel computation. In this paper, we seek to dispense with the windowing scheme and introduce a more elaborate and parallelizable method to exploit locality. To this end, we propose a new MLP module, namely Shifted-Pillars-Concatenation (SPC), that consists of two steps of processes: (1) Pillars-Shift, which generates four neighboring maps by shifting the input image along four directions, and (2) Pillars-Concatenation, which applies linear transformations and concatenation on the maps to aggregate local features. SPC module offers superior local modeling power and performance gains, making it a promising alternative to the convolutional layer. Then, we build a pure-MLP architecture called Caterpillar by replacing the convolutional layer with the SPC module in a hybrid model of sMLPNet [40]. Extensive experiments show Caterpillar's excellent performance on both small-scale and ImageNet-1k classification benchmarks, with remarkable scalability and transfer capability possessed as well. The code is available at *https://github.com/sunjin19126/Caterpillar*.

## CCS Concepts

• **Multimedia Foundation Models → Vision and Language**.

*Corresponding author.

## Keywords

Computer Vision, Pure-MLP Architecture, Caterpillar, SPC Module

**ACM Reference Format:**
Jin Sun, Xiaoshuang Shi, Zhiyuan Wang, Kaidi Xu, Heng Tao Shen, and Xiaofeng Zhu. 2024. Caterpillar: A Pure-MLP Architecture with Shifted-Pillars-Concatenation. In *Proceedings of the 32nd ACM International Conference on Multimedia (MM '24), October 28-November 1, 2024, Melbourne, VIC, Australia.* ACM, New York, NY, USA, 15 pages. https://doi.org/10.1145/3664647.3680809

## 1 Introduction

Deep architectures in computer vision have evolved from Convolutional Neural Networks (CNNs), through Vision Transformers (ViTs), and now to Multi-Layer Perceptrons (MLPs). CNNs [16, 22, 39] primarily utilize *convolution* to aggregate local features but struggle to capture global dependencies between long-range pillars (tokens) in an image. ViTs [9, 44] employ *self-attention mechanism* to consider all pillars from a global perspective. Unfortunately, the *self-attention mechanism* suffers from high computational complexity. To overcome this weakness, MLP-based models [42] replace the self-attention layers with simple MLPs to perform *token(spatial)-mixing* across the input pillars, thereby significantly reducing the computational costs. However, early MLP models [42, 43] encounter the challenges in sufficiently incorporating local dependencies. As a solution, researchers have proposed hybrid models [27, 40] that combine convolutional layers with MLPs to achieve a balance between capturing local and global information, bringing stable performance improvements.

Convolutional layers slide a local window across an image to introduce locality and translation-invariance, which have brought great successes for CNNs [16, 22] and also inspired a number of influential ViTs [33, 52]. Nevertheless, convolution has inherent drawbacks. First, it may introduce redundancy, especially to the edge features. The convolution aggregates pixels in a local window with a larger receptive scope, while the edge features, such as *shape* and *contour*, often consist of only a few pixels that cannot

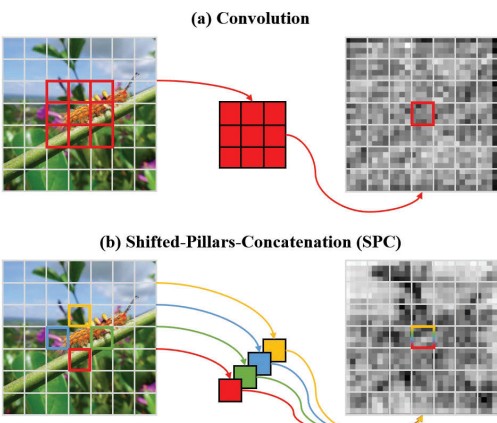

**(a) Convolution**

**(b) Shifted-Pillars-Concatenation (SPC)**

**Figure 1: (a) The convolutional layer sequentially slides a local window across each pillar (token) with a larger receptive field (i.e., the colored border), leading to low parallel computation and redundant representation. (b) The proposed SPC module adopts a window-free strategy. It applies four linear filters which encode the local features for all pillars in parallel from their neighbors of four directions, exploiting the locality elaborately and simultaneously.**

fully fill the scope. Therefore, the edges can get mixed information with the background, leading to redundant representation. Additionally, the sliding window needs to encode features sequentially and individually at each position. This sequential nature leads to convolution calculations with limited parallel computing capability.

In this paper, we seek to break the sequential windowing scheme and present an alternative to convolution. To this end, we propose a new MLP-based module called Shifted-Pillars-Concatenation (SPC), which consists of two processes: Pillars-Shift and Pillars-Concatenation. In Pillars-Shift, we shift the input image along four directions (*i.e.,* up, down, left, right) to create four neighboring maps, with the local information for all pillars decomposed into four respective groups according to the orientation of neighboring pillars. In Pillars-Concatenation, we apply four linear transformations to individually encode these maps of discrete groups and then concatenate them together, with each pillar achieving the simultaneous and elaborate aggregation of local features from its four neighbors. Based on the proposed SPC, we introduce a pure-MLP architecture namely Caterpillar, which is built by replacing the (depth-wise) convolutional layer with the proposed SPC module in a Conv-MLP hybrid model (*i.e.,* sMLPNet[40]). The Caterpillar inherits the advantages of sMLPNet, which clearly separates the *local-* and *global-mixing* operations in its spatial-mixing blocks and utilizes the sparse-MLP (sMLP) layer to aggregate global features (Figure 3, left), while leveraging the SPC module to exploit locality.

For experiments, we uniformly validate the direct application of the Caterpillar with various vision architectures (*i.e.,* CNNs, ViTs, MLPs, Hybrid models) on common-used small-scale images

[21, 47, 53], among which the Caterpillar achieves the best performance on all used datasets. On the popular ImageNet-1K benchmark, the Caterpillar series attains better or comparable performance to recent state-of-the-art methods (*e.g.,* Caterpillar-B, 83.7%). Caterpillar also possesses excellent scalability and transfer capability through corresponding experiments. On the other hand, in all experiments, the Caterpillars obtain higher accuracy than the baseline sMLPNets, while changing the convolution to the SPC module in ResNet-18 brings 4.7% top-1 accuracy gains on ImageNet-1K dataset, demonstrating the potential of SPC to serve as an alternative to convolution in both plug-and-play and independent ways.

In summary, the major contributions of this paper are listed as follows:

- We propose a novel SPC module, which adopts a window-free scheme and can exploit local information more elaborately and simultaneously than convolution.
- We introduce a new pure-MLP model called Caterpillar, which utilizes SPC and sMLP module to aggregate the local and global information in a sequential way.
- Extensive experiments demonstrate the excellent scalability, transfer capability, as well as classification performance of the Caterpillar on both small- and large-scale image recognition tasks, with better performance of SPC than convolution.

## 2 Related Work

### 2.1 Local Modeling Approaches

The idea of local modeling can be traced back to research on the organization of the visual cortex [18, 19], which inspired Fukushima to introduce the Cognitron [10], a neural architecture that models nearby features in local regions. Departing from biological inspiration, Fukushima further proposed Neocognitron [11], which introduces weight sharing across spatial locations through a sliding window strategy. LeCun combined weight sharing with back-propagation algorithm and introduced LeNet [23–25], laying the foundation for the widespread adoption of CNNs in the Deep Learning era. Since 2012, when AlexNet [22] achieved remarkable performance in the ImageNet classification competition, convolution-based methods have dominated the field of computer vision for nearly a decade. With the popularity of CNNs, research efforts have been devoted to improving individual convolutional layers, such as depth-wise convolution [54] and deformable convolution [7]. On the other hand, alternative approaches to replace convolution have also been explored, such as the *shift*-based methods involving sparse-shift [3] and partial-shift [30]. The idea behind these approaches is to move each channel of the input image in different spatial directions, and mix spatial information through linear transformations across channels. The proposed SPC module also builds upon the *shift* idea but shifts the entire image into four neighboring maps in the process of Pillars-Shift, while making use of the linear projections and concatenation in Pillars-Concatenation.

### 2.2 Neural Architectures for Vision

**CNNs and Vision Transformers.** CNNs have achieved remarkable success in computer vision, with well-known models including

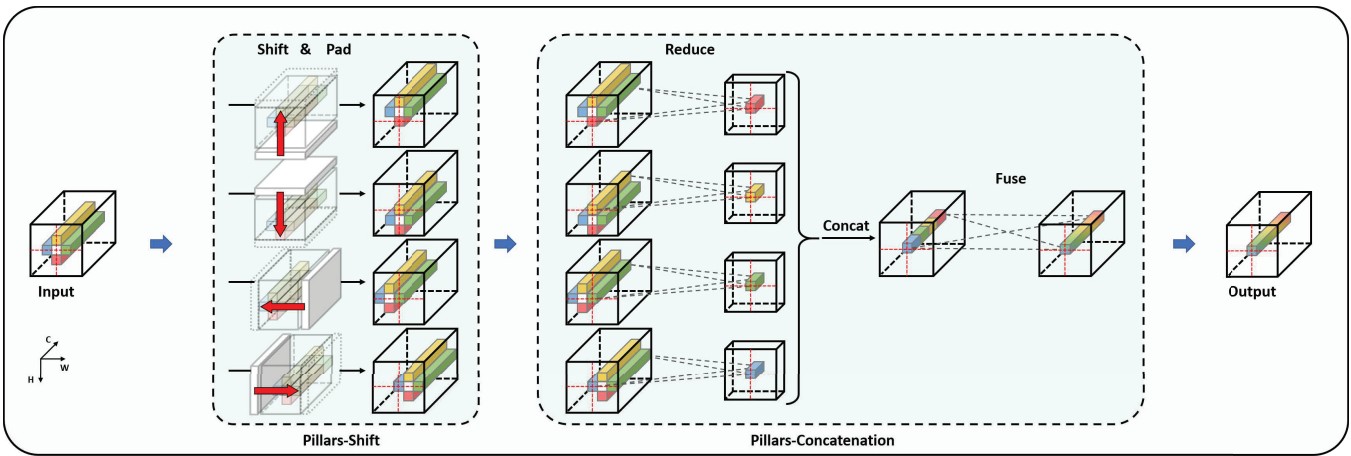

**Figure 2: The SPC module consists of two processes: Pillars-Shift (*Shift + Pad*) and Pillars-Concatenation (*Reduce + Concat + Fuse*). In Pillars-Shift, the input image is recurrently shifted along four directions to create neighboring maps, while *Pad* is used to maintain the feature size by padding these maps with pillars of specific values. In Pillars-Concatenation, *Reduce* is achieved through four C × C/4 linear projections, and *Fuse* is accomplished through a C × C linear projection, where C represents the number of input feature channels.**

AlexNet [22], VGG [39] and ResNet [16]. The attention-based Transformer, initially proposed for machine translation [46], has been successfully applied to vision tasks with the introduction of Vision Transformer (ViT) [9]. Since then, various advancements have been proposed to improve training efficiency and model performance for ViTs, such as data-efficient training strategy [44] and pyramid architecture [33, 49], which have also benefited the entire vision field. At the core of Transformer models lies the *multi-head self-attention mechanism*. The proposed SPC module shares the similar operation to the *multi-head* settings, as it encodes local neighboring information from different representation subspaces with multiple linear filters in the Pillars-Concatenation process.

**Vision MLPs.** Vision MLPs [2, 12, 17, 29, 41–43, 50, 55, 56] have also made significant progress since the invention of MLP-Mixer [42], which alternatively conducts the *token-mixing* (cross-location) operations and *channel-mixing* (per-location) operations to aggregate spatial and channel information, respectively. Early MLP-based models, such as MLP-Mixer [42] and ResMLP [43], perform *token-mixing* across all pillars from a global perspective, lacking the ability to effectively model local features. As a result, a number of studies propose to enhance MLPs with local modeling capabilities. Hire-MLP [12], for instance, performs *Inner-* and *Cross-region Rearrangement* to encode the local and global information in parallel, while AS-MLP [29] adopts an *axial-shift strategy* that shifts each channel of the image along two directions. Our Caterpillar is built with the SPC module, which shifts the entire image into four neighboring maps, enabling the elaborate and simultaneous encoding of local information for all pillars.

**Hybrid Architectures.** Apart from the pure-MLP methods[12], which capture both local and global dependencies fully in MLP-based approaches, there have been developments in combining MLPs with convolutional layers to separately aggregate these two

types of information [1, 27, 40]. Among them, sMLPNet [40] introduces a *sparse-MLP module* to aggregate global information while using the *depth-wise convolutional (DWConv) layer* to model local features. Concurrent with our work, Strip-MLP [1] chiefly replaces the *sparse-MLP* (*i.e.,* the global-mixing module) in sMLPNet with a *Strip-MLP layer*, achieving superior scores on both large- and small-scale image datasets. The proposed Caterpillar is also built upon the sMLPNet but replaces the *DWConv* (*i.e.,* the local-mixing module) with the SPC module, resulting in a pure-MLP architecture, which also attains excellent performance on various-scale image recognition tasks.

## 3 Method

### 3.1 Shifted-Pillars-Concatenation Module

In this section, we first introduce the SPC module, of which the working procedure can be decomposed into Pillars-Shift and Pillars-Concatenation, as shown in Figure 2. Then, we analyze its computational parameters with that of the standard and depth-wise convolutional layers.

#### 3.1.1 Shifted-Pillars-Concatenation
**Pillars-Shift.** This process is to shift and pad an input image into four neighboring maps, which can be formulated as:

$$\text{PS}\left(\mathbf{X} \mid dir, s, p_m\right) = \text{Pad}\left(\text{Shift}\left(\mathbf{X}, dir, s\right), p_m\right), \quad dir \subseteq \mathcal{D}_s, \quad (1)$$

where $\mathbf{X}$ is an image, $dir$, $s$ and $p_m$ denote the shifting direction, shifting steps and padding mode, respectively. $\mathcal{D}_s$ is a set containing shifting directions.

Specifically, let $\mathbf{x}_{ij} \in \mathbb{R}^C$ denote a feature vector (referred to as "pillar" and also depicted as pillars in Figure 2, so as to clearly and accurately express and visualize the workflow in the SPC module),

we can have an image:

$$\mathbf{X}_{in} = \begin{bmatrix} \mathbf{x}_{11} & \mathbf{x}_{12} & \cdots & \mathbf{x}_{1W} \\ \mathbf{x}_{21} & \mathbf{x}_{22} & \cdots & \mathbf{x}_{2W} \\ \vdots & \vdots & \vdots & \vdots \\ \mathbf{x}_{(H-1)1} & \mathbf{x}_{(H-1)2} & \cdots & \mathbf{x}_{(H-1)W} \\ \mathbf{x}_{H1} & \mathbf{x}_{H2} & \cdots & \mathbf{x}_{HW} \end{bmatrix},$$

which means $\mathbf{X}_{in} \in \mathbb{R}^{H \times W \times C}$, where $H$, $W$ and $C$ represent the width, height and channel number, respectively. Taking $\mathbf{X}_u$ (*i.e.,* the up-wise neighboring map) as an example, we first perform a *shift* operation on $\mathbf{X}_{in}$ by setting $dir=$'$up'$ and $s = 1$, so that $\mathbf{X}_{in}$ is transformed to:

$$\mathbf{X}'_u = \begin{bmatrix} \mathbf{x}_{21} & \mathbf{x}_{22} & \cdots & \mathbf{x}_{2W} \\ \vdots & \vdots & \vdots & \vdots \\ \mathbf{x}_{(H-1)1} & \mathbf{x}_{(H-1)2} & \cdots & \mathbf{x}_{(H-1)W} \\ \mathbf{x}_{H1} & \mathbf{x}_{H2} & \cdots & \mathbf{x}_{HW} \end{bmatrix},$$

where $\mathbf{X}'_u \in \mathbb{R}^{(H-1) \times W \times C}$. Then, we *pad* $\mathbf{X}'_u$ according to the Zero Padding and attain the $\mathbf{X}_u$:

$$\mathbf{X}_u = \begin{bmatrix} \mathbf{x}_{21} & \mathbf{x}_{22} & \cdots & \mathbf{x}_{2W} \\ \vdots & \vdots & \vdots & \\ \mathbf{x}_{(H-1)1} & \mathbf{x}_{(H-1)2} & \cdots & \mathbf{x}_{(H-1)W} \\ \mathbf{x}_{H1} & \mathbf{x}_{H2} & \cdots & \mathbf{x}_{HW} \\ 0 & 0 & \cdots & 0 \end{bmatrix},$$

where $\mathbf{X}_u \in \mathbb{R}^{H \times W \times C}$. By default settings with $\mathcal{D}_s$ of ['$up'$, '$down'$, '$left'$, '$right'$], the input $\mathbf{X}_{in}$ will be transformed into four neighboring maps of $\mathbf{X}_u, \mathbf{X}_d, \mathbf{X}_l, \mathbf{X}_r$, where $\mathbf{X}_d, \mathbf{X}_l, \mathbf{X}_r \in \mathbb{R}^{H \times W \times C}$.

**Pillars-Concatenation.** Obviously, Pillars-Shift has no parameter learning, which would weaken the representation capability of the module. To overcome this deficiency, we introduce the Pillars-Concatenation process. Specifically, the neighboring maps $\mathbf{X}_u, \mathbf{X}_d, \mathbf{X}_l, \mathbf{X}_r$ are projected through four independent fully-connected (FC) layers. The parameters are $\mathbf{W}_u, \mathbf{W}_d, \mathbf{W}_l, \mathbf{W}_r \in \mathbb{R}^{C \times C/4}$, respectively, so as to reduce the number of neighboring maps' channels into $C/4$. After that, all of the reduced maps are concatenated along the channel dimension and then projected again, by an FC layer with the parameters $\mathbf{W} \in \mathbb{R}^{C \times C}$ to fuse the local features. This process can be represented as:

$$\text{PC}(\mathbf{X}) = \text{Concat}(\mathbf{X}_u \mathbf{W}_u, \mathbf{X}_d \mathbf{W}_d, \mathbf{X}_l \mathbf{W}_l, \mathbf{X}_r \mathbf{W}_r) \mathbf{W}, \quad (2)$$

Through the Pillars-Concatenation, the four neighboring maps are reduced, concatenated and finally fused into the $\mathbf{X}_{out} \in \mathbb{R}^{H \times W \times C}$, with the local information for all pillars within the image aggregated in parallel.

### 3.1.2 Parameter Analysis with Convolutions

For an input image with the input dimension of $d_{in}$ and output dimension of $d_{out}$, the number of parameters in standard and depth-wise convolution (DWConv) can be calculated as $d_{in} \times k^2 \times d_{out}$ and $d_{in} \times k^2$, respectively, with $k$ representing the kernel size. In comparison, The parameters of SPC module are $d_{in} \times d_{in} + d_{in} \times d_{out}$ (detailed as $d_{in} \times d_{in}/4 \times 4 + d_{in} \times d_{out}$). In the typical scenario, where

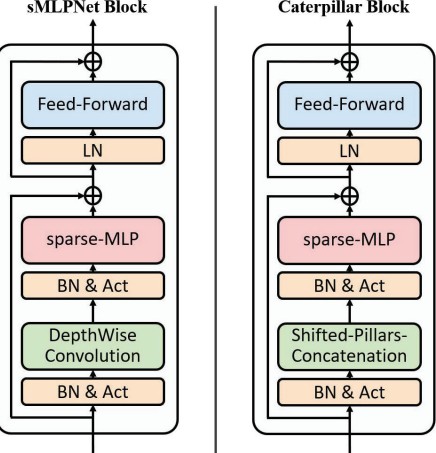

**Figure 3: The structures of sMLPNet and Caterpillar blocks.**

$d_{in}$ is equal to $d_{out}$, the parameters of a standard $3 \times 3$ convolution are $9 \times d_{in}^2$, which is 4.5 times larger than that of SPC, *i.e.,* $2 \times d_{in}^2$, demonstrating that the SPC module has lower computational complexity than the standard convolutional layers. Additionally, the depth-wise settings reduce the parameters in DWConv into $9 \times d_{in}$, which is lower than SPC and might inspired future works that improve SPC through such lightweight techniques.

### 3.2 Caterpillar Block

Caterpillar block is built by replacing the depth-wise convolution with the SPC module in sMLPNet block [40], in which a sparse-MLP (sMLP) module (illustrated in Appendix A.8) is introduced for aggregating global features. As illustrated in Figure 3, a Caterpillar block contains three basic modules: an SPC module and an sMLP module, with a BatchNorm (BN) and a GELU nonlinearity applied before them, and a FFN module, which follows a LayerNorm (LN) layer. The SPC and sMLP form the token-mixing component and the FFN servers as channel-mixing module, with applied two residual connections.

Given an image $\mathbf{X} \in \mathbb{R}^{H \times W \times C}$, the calculation in the Caterpillar block can be formulated as:

$$\mathbf{X}' = \text{SPC}(\text{GELU}(\text{BN}(\mathbf{X}))), \quad (3)$$

$$\mathbf{Y} = \text{sMLP}(\text{GELU}(\text{BN}(\mathbf{X}'))) + \mathbf{X}, \quad (4)$$

$$\mathbf{Z} = \text{FFN}(\text{LN}(\mathbf{Y})) + \mathbf{Y}, \quad (5)$$

where $\mathbf{X}'$ denotes the output features of the SPC layer, $\mathbf{Y}$ and $\mathbf{Z}$ represent the output of token-mixing and channel-mixing modules, respectively.

### 3.3 Caterpillar Architectures

We build the Caterpillar architectures in a pyramid structure of four stages, which first represent the input images into patch-level features, and gradually shrink the spatial size of the feature maps as the network deepens. This enables Caterpillar to leverage the scale-invariant property of images as well as make full use of multi-scale features.

We introduce five variants of the Caterpillar architectures, *i.e.*, Caterpillar-Mi, -Tx, -T, -S and -B, with different numbers of Caterpillar blocks stacked in their four stages. The architecture hyperparameters of these variants are:

- Caterpillar-Mi: $C = 40$, layer numbers = {2,6,10,2}
- Caterpillar-Tx: $C = 60$, layer numbers = {2,8,14,2}
- Caterpillar-T: $C = 80$, layer numbers = {2,8,14,2}
- Caterpillar-S: $C = 96$, layer numbers = {2,10,24,2}
- Caterpillar-B: $C = 112$, layer numbers = {2,10,24,2}

where $C$ is the channel number of the hidden layers in the first stage, and layer numbers denote the number of blocks in each of their four stages. Detailed configurations are in Appendix A .5.

## 4    Experiments

We compare the classification performance of Caterpillar with various vision models on small-scale images as well as ImageNet-1k dataset in Sec. 4 .1, 4 .5 and Sec. 4 .2, respectively, with its scalability and transfer capability also verified in Sec. 4 .1 and Sec. 4 .5. Then, we conduct ablation studies in Sec. 4 .3 to find the best settings for the proposed SPC module and Caterpillar architecture. Additionally, the comparison between sMLPNet and Caterpillar in Sec. 4 .1, 4 .2 as well as the experiments in Sec. 4 .6 is to verify SPC to be an alternative to convolution in both plug-and-play and independent manners, with visualization analysis for SPC provided in Sec. 4 .4.

### 4 .1    Small-scale Image Classification

**Datasets.** We conduct small-scale image recognition experiments on four commonly-used benchmarks: Mini-ImageNet (MIN) [47], CIFAR-10 (C10) [21], CIFAR-100 (C100) [21] and Fashion-MNIST (Fashion) [53]. We utilize these images in their original sizes, different from the settings in [1] that resized images into $224 \times 224$.

**Experimental Settings.** We evaluate the Caterpillar with fourteen representative vision models, including six MLP models [2, 12, 17, 41, 43, 50], two CNN models [16, 45], three Transformer models [33, 44, 60], and three hybrid models [1, 14, 40], as tagged in Table 1. All models are directly trained on the small images without extra data. To enable the models adaptable to small-sized images (*e.g.*, 32 × 32), we change their *patch embedding layers* into small *patch sizes* according to uniform rules, of which the detailed implementation is provided in Appendix A .6. For a fair comparison, we adopt the same training strategies that were presented in their original papers (for ImageNet-1K), which are presented in Appendix A .7.

**Results.** Table 1 presents the classification results of different methods on the four small-scale image datasets. As we can see, the proposed Caterpillar outperforms the sMLPNet on all the four benchmarks (*e.g.*, Caterpillar-T$^\dagger$, 23M, 77.56% *vs* sMLPNet-T, 24M, 77.07% on MIN), showcasing the better classification capability of the SPC than convolutional layers and its potential to be an alternative to convolution in plug-and-play ways. Additionally, the Caterpillar attain the best scores among all tested architectures, *i.e.*, the Caterpillar-T reaches 78.16% accuracy on MIN, 97.10% on C10, 84.86% on C100, and 95.72% on Fashion, making it an effective tool for small-scale image recognition tasks.

**Scalability analysis.** "Simple algorithms that scale well are the core of deep learning" [15]. Thus, we scale the Caterpillar from -Mi with FLOPs about 0.4G to -B about 5.5G, *i.e.*, Caterpillar-Mi,

**Table 1: Results (%) of Caterpillar and other MLP / CNN / Transformer / Hybrid vision models on four small-scale datasets. As the model parameters and FLOPs are similar on these datasets, we just report those metrics on CIFAR-10 for clarity. The Caterpillar-T$^\dagger$ scales the number of channels to [72, 144, 288, 576], with similar computational costs to the sMLPNet-T. ▲ CNN, ♦ Transformer, ■ MLP, ■ Hybrid, ★ Ours.**

| Networks | MIN | C10 | C100 | Fashion | Params | FLOPs |
|---|---|---|---|---|---|---|
| ♦ DeiT-Ti[44] | 54.55 | 88.87 | 67.46 | 92.97 | 5.4M | 0.3G |
| ♦ NesT-T[60] | 73.44 | 94.05 | 75.60 | 94.26 | 6.4M | 2.3G |
| ■ CCT-7/3×1[14] | – | 91.80 | 74.09 | 93.70 | 3.7M | 0.9G |
| ★ **Caterpillar-Mi** | **74.14** | **95.54** | **79.41** | **95.14** | 5.9M | 0.4G |
| ▲ ResNet-18[43] | 70.95 | 95.54 | 77.66 | 95.11 | 11.2M | 0.7G |
| ▲ ConvMixer_768/32[45] | 57.94 | 91.54 | 70.13 | 93.36 | 19.4M | 1.2G |
| ■ ResMLP-S12[43] | 68.63 | 93.67 | 76.44 | 94.58 | 14.3M | 0.9G |
| ■ CycleMLP-B1[2] | 70.68 | 88.06 | 66.17 | 92.87 | 12.7M | 0.1G |
| ■ HireMLP-Tiny[12] | 71.66 | 86.42 | 62.13 | 92.35 | 17.6M | 0.1G |
| ■ Wave-MLP-T[41] | 72.15 | 88.85 | 65.92 | 92.83 | 16.7M | 0.1G |
| ■ Strip-MLP-T*[1] | 76.05 | 96.34 | 82.53 | 95.33 | 16.3M | 0.8G |
| ★ **Caterpillar-Tx** | **77.27** | **96.54** | **82.69** | **95.38** | 16.0M | 1.1G |
| ▲ ResNet-34[16] | 72.03 | 95.92 | 79.53 | 95.48 | 21.3M | 1.5G |
| ▲ ResNet-50[16] | 72.65 | 96.06 | 79.11 | 95.28 | 23.7M | 1.6G |
| ♦ DeiT-S[44] | 42.41 | 83.10 | 64.65 | 93.43 | 21.4M | 1.4G |
| ♦ Swin-T[33] | 53.11 | 85.69 | 67.60 | 89.90 | 27.6M | 1.4G |
| ■ ResMLP-S24[43] | 69.63 | 94.76 | 78.65 | 95.27 | 28.5M | 1.9G |
| ■ CycleMLP-B2[2] | 71.11 | 88.84 | 67.83 | 93.41 | 22.6M | 0.1G |
| ■ HireMLP-Small[12] | 73.86 | 88.51 | 62.54 | 92.70 | 32.6M | 0.1G |
| ■ Wave-MLP-S[41] | 67.51 | 88.37 | 63.24 | 92.96 | 30.2M | 0.1G |
| ■ ViP-Small/7[17] | 70.94 | 94.12 | 78.28 | 95.22 | 24.7M | 1.7G |
| ■ DynaMixer-S[50] | 71.40 | 95.32 | 78.34 | 95.14 | 25.2M | 1.8G |
| ■ sMLPNet-T[40] | 77.07 | 96.87 | 82.89 | 95.53 | 23.5M | 1.6G |
| ■ Strip-MLP-T[1] | 76.47 | 96.48 | 82.59 | 95.50 | 22.5M | 1.2G |
| ★ **Caterpillar-T$^\dagger$** | 77.56 | 97.08 | 83.12 | 95.57 | 23.0M | 1.6G |
| ★ **Caterpillar-T** | **78.16** | **97.10** | **83.86** | **95.72** | 28.4M | 1.9G |
| ★ **Caterpillar-S** | **78.94** | **97.22** | **84.40** | **95.80** | 58.0M | 4.1G |
| ★ **Caterpillar-B** | **79.06** | **97.35** | **84.77** | **95.85** | 78.8M | 5.5G |

-Tx, -T, -S and -B. It is credible that Caterpillar exhibits excellent scalability on small-scale datasets, as it obtains steady improvement from bigger models.

### 4 .2    ImageNet Classification

**Datasets.** We test the Caterpillar on the ImageNet-1K benchmark [8], which consists of 1.28M training and 50K validation images belonging to 1,000 categories.

**Experimental Settings.** We train our models on 8 NVIDIA GeForce RTX 3090 GPUs with gradient accumulation techniques. For training strategies, we employ the AdamW [35] optimizer to train our models for 300 epochs, with a weight decay of 0.05 and a batch size of 1024. The learning rate is initially 1e-3 and gradually drops to 1e-5 according to the consine schedule. The data augmentation includes Random Augment [6], Mixup [58], Cutmix [57], Random Erasing [61]. More details are shown in Appendix A .7.

**Results.** Table 2 presents the performance of Caterpillar with other well-established methods on the ImageNet-1k benchmark. Similar to the results in Section 4 .1, the Caterpillar models consistently

**Table 2: Results (%) of Caterpillar and other vision models on ImageNet-1K datasets. ▲ CNN, ♦ Transformer, ■ MLP, ▨ Hybrid, ★ Ours.**

| Networks | Params | FLOPs | Top-1 |
|---|---|---|---|
| ♦ DeiT-Ti[44] | 5M | 1.1G | 72.2 |
| ■ gMLP-Ti[31] | 6M | 1.4G | 72.3 |
| ★ **Caterpillar-Mi** | 6M | 1.2G | **76.3** |
| ▲ ResNet-18[16, 51] | 12M | 1.8G | 70.6 |
| ■ ResMLP-S12[43] | 15M | 3.0G | 76.6 |
| ■ Hire-MLP-Ti[12] | 18M | 2.1G | 79.7 |
| ■ Wave-MLP-T[41] | 17M | 2.4G | 80.6 |
| ▨ Strip-MLP-T*[1] | 18M | 2.5G | **81.2** |
| ★ **Caterpillar-Tx** | 16M | 3.4G | 80.9 |
| ▲ ResNet-50[16, 51] | 26M | 4.1G | 79.8 |
| ▲ RegNetY-4G[37] | 21M | 4.0G | 80.0 |
| ♦ DeiT-S[44] | 22M | 4.6G | 79.8 |
| ♦ Swin-T[33] | 29M | 4.5G | 81.3 |
| ■ ResMLP-S24[43] | 30M | 6.0G | 79.4 |
| ■ ViP-Small/7[17] | 25M | 6.9G | 81.5 |
| ■ AS-MLP-T[29] | 28M | 4.4G | 81.3 |
| ■ Hire-MLP-S[12] | 33M | 4.2G | 82.1 |
| ■ Wave-MLP-S[41] | 30M | 4.5G | **82.6** |
| ▨ sMLPNet-T[40] | 24M | 5.0G | 81.9 |
| ▨ Strip-MLP-T[1] | 25M | 3.7G | 82.2 |
| ★ **Caterpillar-T** | 29M | 6.0G | 82.4 |
| ▲ ResNet-101[16, 51] | 45M | 7.9G | 81.3 |
| ▲ RegNetY-8G[37] | 39M | 8.0G | 81.7 |
| ♦ Swin-S[33] | 50M | 8.7G | 83.0 |
| ■ ViP-Medium/7[17] | 55M | 16.3G | 82.7 |
| ■ AS-MLP-S[29] | 50M | 8.5G | 83.1 |
| ■ Hire-MLP-B[12] | 58M | 8.1G | 83.2 |
| ■ Wave-MLP-M[41] | 44M | 7.9G | 83.4 |
| ▨ sMLPNet-S[40] | 49M | 10.3G | 83.1 |
| ▨ Strip-MLP-S[1] | 44M | 6.8G | 83.3 |
| ★ **Caterpillar-S** | 60M | 12.5G | **83.5** |
| ▲ ResNet-152[16, 51] | 60M | 11.6G | 81.8 |
| ▲ RegNetY-16G[37] | 84M | 16.0G | 82.9 |
| ♦ DeiT-B[44] | 86M | 17.5G | 81.8 |
| ♦ Swin-B[33] | 88M | 15.4G | 83.5 |
| ■ ResMLP-B24[43] | 116M | 23.0G | 81.0 |
| ■ ViP-Large/7[17] | 88M | 24.4G | 83.2 |
| ■ AS-MLP-B[12] | 88M | 15.2G | 83.3 |
| ■ Hire-MLP-B[12] | 96M | 13.4G | **83.8** |
| ■ Wave-MLP-B[41] | 63M | 10.2G | 83.6 |
| ▨ sMLPNet-B[40] | 66M | 14.0G | 83.4 |
| ▨ Strip-MLP-B[1] | 57M | 9.2G | 83.6 |
| ★ **Caterpillar-B** | 80M | 17.0G | 83.7 |

outperform their sMLPNet counterparts, which further emphasizes the superiority of the SPC module over convolutional layers, highlighting its potential as a plug-and-play replacement to convolution. Furthermore, the Caterpillar series exhibit competitive or even superior performance to state-of-the-art networks. For instance, Caterpillar-B achieves the top-1 accuracy of 83.7%, which slightly surpasses several representative MLP architectures (*e.g.,* Wave-MLP-B, 83.6%, AS-MLP-B, 83.3%, ViP-Large/7, 83.2%),

verifying the efficacy of Caterpillar in tackling large-scale vision recognition tasks.

## 4 .3 Ablation Study

In this section, we ablate essential design components in the proposed Caterpillar architecture. We use the same datasets and experimental settings as in Section 4 .1. The base architecture is Caterpillar-T.

### 4.3.1 Pillars-Shift of SPC

**Number of shift directions.** This hyper-parameter ($N_D$) controls the shifting directions of input images so as to determine the receptive field of SPC on local features. We experiment with $N_D$ values ranging from 4 to 9. Among them, 4 represents the scope of four neighboring directions (*up, down, left, and right*), and 5 includes the *center* pillar itself. 8 covers a wider scope, including *up, down, left, right, up-left, up-right, down-left, and down-right* directions. When $N_D$ is set to 9, it adds the *center* pillar itself, which is similar to the scope of 3x3 convolution. From Table 3, increasing the number of neighbors can bring redundancy, because more background information can be injected into the target pillar – the similar drawback existed in convolution. This result underscores that the local features can be sufficiently obtained from a 4-scoped receptive field.

**Table 3: Results (%) on different numbers of shift directions in the Pillars-Shift process. The $C$ for the "$N = 9$" is adjusted to [81, 162, 324, 648] to ensure the divisibility of $C$ by $N$ with the *Reduce Weights* of $W \in \mathbb{R}^{C \times C/N}$**

| Num. of dir. ($N_D$) | MIN | C10 | C100 | Fashion | Params | FLOPs |
|---|---|---|---|---|---|---|
| 4 | 78.16 | **97.10** | **83.86** | **95.72** | 28.4M | 1.9G |
| 5 | 78.04 | 96.93 | 83.55 | 95.63 | 28.4M | 1.9G |
| 8 | **78.19** | 96.92 | 83.58 | 95.57 | 28.4M | 1.9G |
| 9* | 77.92 | 96.82 | 83.60 | 95.52 | 29.1M | 2.0G |

**Number of shift steps.** The hyper-parameter $s$ determines the range of local features for the Pillars-Shift operation. When $s$ is set to 0, 1, or 2, the input image is shifted 0, 1, or 2 steps along the corresponding directions, which allows the local information for each pillar to be aggregated from itself (no shifting, lacking local modeling capability), neighboring pillars (with a distance of 1), or distant pillars (with a distance of 2), respectively. Table 4 displays the results of the proposed method with different numbers of shifting steps. The findings indicate that the best performance can be achieved when $s = 1$.

**Table 4: The model accuracy (%) on three different shift steps in the Pillars-Shift process.**

| shift steps ($s$) | MIN | C10 | C100 | Fashion |
|---|---|---|---|---|
| 0 | 76.71 | 95.84 | 81.12 | 95.22 |
| 1 | **78.16** | **97.10** | **83.86** | **95.72** |
| 2 | 76.29 | 96.17 | 82.04 | 95.26 |

**Type of padding modes.** The padding operation is to supplement pillars to the tail in neighboring maps. Different modes can decide

what noises (extra information) would be injected to the pillars on the margin of images. We test four popular padding modes in Table 5. Among them, *Replicated Padding* can inject repeated features to the marginal pillars, which might bring redundancy. Both the *Circular* and *Reflect* modes add long-distance information to those pillars, which is obviously detrimental to the locality bias. *Zero Padding* is clean and thus achieves the best accuracy.

**Table 5: The model accuracy (%) on four different padding modes in the Pillars-Shift process.**

| Padding Mode ($p_m$) | MIN | C10 | C100 | Fashion |
|---|---|---|---|---|
| Zero | **78.16** | **97.10** | **83.86** | **95.72** |
| Replicated | 78.13 | 96.88 | 83.35 | 95.41 |
| Circular | 78.08 | 96.81 | 83.40 | 95.45 |
| Reflect | 77.76 | 96.89 | 83.33 | 95.40 |

**4.3.2 Pillars-Concatenation of SPC**

The Pillars-Concatenation process involves three key operations: (1) *Reduce*, which enables diverse representation by transforming neighboring maps in multiple representation spaces; (2) *Concatenation (Concat)*, which integrates four neighboring maps to combine local features for all pillars in parallel; and (3) *Fuse*, which selectively learns and weights neighboring features to enhance the representation capabilities. We ablate five combinations of these operations in Table 6 and observe that *Reduce+Concat+Fuse* and *Concat+Fuse* perform better than the other options. Among them, *Reduce+Concat+Fuse* achieves a better trade-off between computational costs and accuracy.

**Table 6: Results (%) of different ways to mix local neighbors.**

| Mixing ways | MIN | C10 | C100 | Fashion | Params | FLOPs |
|---|---|---|---|---|---|---|
| Reduce+Concat+Fuse | 78.16 | **97.10** | 83.86 | **95.72** | 28.4M | 1.9G |
| Reduce+Concat | 77.72 | 97.02 | 83.26 | **95.72** | 25.9M | 1.8G |
| Concat+Fuse | **78.37** | 97.06 | **83.94** | 95.45 | 33.3M | 2.3G |
| Sum+Fuse | 76.14 | 96.16 | 80.88 | 95.28 | 25.9M | 1.8G |
| Sum | 74.81 | 95.33 | 75.48 | 95.34 | 23.4M | 1.6G |

**4.3.3 Local-Global Combination Strategies**

In Section 4 .1 and 4 .2, the sMLPNet, Strip-MLP, and Caterpillar all attained excellent performance on both small- and large-scale image recognition tasks. This success could be attributed to the common strategy that they clearly separate the local- and global-mixing operations in their token-mixing modules. However, both the sMLPNet and Strip-MLP missed the experiments to further explore the effects of local-global combination ways. To fill this gap, we conduct this ablation and perform various strategies as depicted in Figure 4. We evaluate six strategies, denoted as (a), (b), (c) for sequential regimes, and (e), (f), (g) for parallel methods, by rearranging the SPC and sMLP modules in Caterpillar blocks. Table 7 shows that the simply sequential methods generally outperform the complicated parallel strategies. We attribute this phenomenon to the idea that the 'High-Cohesion and Low-Coupling' principle leads to higher performance, since the internal SPC and sMLP modules are both working in sophisticated parallel ways. Furthermore, the L-G strategy achieves the best performance.

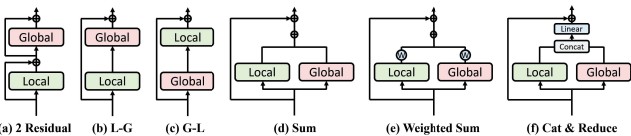

**(a) 2 Residual  (b) L-G  (c) G-L  (d) Sum  (e) Weighted Sum  (f) Cat & Reduce**

**Figure 4: Different ways to combine local and global information.**

**Table 7: Comparison (%) between six different strategies for combining local and global information.**

| Combine ways | MIN | C10 | C100 | Fashion | Params | FLOPs |
|---|---|---|---|---|---|---|
| 2 Residual | 77.06 | 96.92 | 82.51 | 95.64 | 28.4M | 1.9G |
| L-G (default) | **78.16** | **97.10** | **83.86** | **95.72** | 28.4M | 1.9G |
| G-L | 78.09 | 96.88 | 83.45 | 95.65 | 28.4M | 1.9G |
| Sum | 76.91 | 96.70 | 82.13 | 95.53 | 28.4M | 1.9G |
| Weighted Sum | 77.94 | 96.82 | 82.56 | 95.60 | 30.3M | 2.0G |
| Concat+Reduce | 76.77 | 96.18 | 82.15 | 95.49 | 33.4M | 2.3G |

## 4 .4 Visualization

To understand how the SPC module processes image data, we visualize the feature maps encoded by SPC coupled with two control ways. Specifically, we build three Caterpillar-T models with the local modules of identity, convolution and SPC, and implement them on the CIFAR-100 dataset. Figure 5 (in Appendix A .1) illustrates the feature maps of six samples, each of which is presented with 3 rows and 4 columns, where rows represent different local-mixing ways and columns are feature maps of different phases in models. For these samples, with the *(a) cattle* as an example, the patterns in SPC features are closer to the convolution and different from the identity. Since convolutional layers capably capture local features, the SPC is also capable of aggregating local information. Furthermore, compared to convolution, the objects in SPC maps exhibit more prominent edge features and are closer to the original input image, indicating that the proposed SPC module can encode local information more elaborately and avoid redundancy issues.

## 4 .5 Analysis with Transfer Learning

### 4.5.1 Transfer Learning Performance of Caterpillar

In this subsection, we compare the Transfer Learning capability of the proposed Caterpillar architecture with recent SOTA models. Following the recent MLP-based models [17, 50], we pre-train the Caterpillar-T on the ImageNet-1k and then fine-tune it on CIFAR-10 and CIFAR-100. From Table 8, the Caterpillar attains higher scores than other representative networks with similar computational costs, indicating that Caterpillar can work well on Transfer tasks.

### 4.5.2 Comparison between Direct and Transfer Strategies

Despite the remarkable Transfer capability, we highlight that Caterpillar can achieve exceptional performance on small-scale images using the 'Direct Training' (Direct) strategy (Section 4 .1). To further illustrate Caterpillar's effectiveness in data-hungry domains without relying on pre-training data, which always faces challenges to domain-shift and task-compatibility, we conduct this study.

**Table 8: The transfer learning results of models pre-trained on ImageNet-1k and fine-tuned to CIFAR-10 and CIFAR-100.**

| Networks | Datasets | Params | Top-1(%) |
|---|---|---|---|
| ViT-S/16 [9] | | 49M | 97.1 |
| ViP-S/7 [17] | CIFAR-10 | 25M | 98.0 |
| DynaMixer-S [50] | | 26M | 98.2 |
| Caterpillar-T | | 29M | **98.3** |
| ViT-S/16 [9] | | 49M | 87.1 |
| ViP-S/7 [17] | CIFAR-100 | 25M | 88.4 |
| DynaMixer-S [50] | | 26M | 88.6 |
| Caterpillar-T | | 29M | **89.3** |

We adopt the same datasets as in Section 4 .1, *i.e.,* MIN, C10, C100 and Fashion, while including two more datasets in certain scientific fields of remote sensing, *i.e.,* Resisc45 (R45) [5] with 27,000 training and 4,500 testing images in 45 categories, and disease diagnosis [20] *i.e.,* Chest_Xray (Chest) with 5,216 training images and 624 testing images belonging to 2 classes. All images are resized to $224 \times 224$. Then, we utilize two Caterpillar-T models as the base architectures. The model of 'Transfer' is pre-trained on the ImageNet-1K and then fine-tune on the target datasets, while the other one for 'Direct' is trained from scratch. From Table 9, the 'Transfer' strategy performs better on MIN, C10, and C100, while the 'Direct' strategy achieves higher scores on Fashion, R45, and Chest, which have dissimilar distributions to the pre-trained data (ImageNet-1K). Therefore, for the data-hungry scientific tasks (especially those without the same distribution to the pre-trained natural images), directly training the deep model can be a more cost-effective approach than the 'Transfer' strategy, with Caterpillar serving as the backbone architecture.

**Table 9: Comparison (%) between 'Transfer Learning' and 'Direct Training' strategies on six small-scale datasets.**

| Networks | Strategy | Epochs | MIN | C10 | C100 | Fashion | R45 | Chest |
|---|---|---|---|---|---|---|---|---|
| Caterpillar-T | Transfer | 300+30 | **95.14** | **98.31** | **89.30** | 95.57 | 97.27 | 93.97 |
| | Direct | 0+300 | 86.98 | 97.60 | 84.67 | **96.13** | **97.35** | **94.29** |

## 4 .6 Exploration for SPC

Previous comparison between Caterpillars and sMLPNets demonstrates the potential of SPC as an alternative to convolution in plug-and-play ways (Table 1, 2). We further explore the SPC to serve as the main module for neural architectures.

**Datasets.** We utilize the same large-scale benchmark of ImageNet-1K as well as the small-scale datasets of MIN, C10, C100 and Fashion, as those in Section 4 .1 and 4 .2.

**Experimental Settings.** We adopt classic ResNet-18 (Res-18) [16] as the baseline CNN. Then, we replace all convolutional layers within Res-18's basic blocks with the SPC module and obtain three SPC-based variants referred to as 'Res-18(SPC)', with $N_C$ utilized as channel numbers to adjust model complexity. For training these models, we follow the 'Procedure A2' in [51].

**Results.** Table 10 displays the ImageNet-1K classification results for the original Res-18 and Res-18(SPC) variants. As we can see, the proposed SPC module can provide higher performance than

**Table 10: Results (%) of Res-18 and Res-18(SPC) on ImageNet-1K. $N_C$ is the channel number of hidden layers in first stage.**

| Networks | $N_C$ | Params | FLOPs | Top-1 (%) |
|---|---|---|---|---|
| Res-18[51] | 64 | 12M | 1.8G | 70.6 |
| | 64 | 3M | 0.6G | 69.1 |
| Res-18(SPC) | 96 | 7M | 1.3G | 73.6 |
| | 128 | 11M | 2.2G | **75.3** |

**Table 11: Results (%) of Res-18 and Res-18(SPC) on four small-scale datasets**

| Networks | $N_C$ | MIN | C10 | C100 | Fashion | Params | FLOPs |
|---|---|---|---|---|---|---|---|
| Res-18[51] | 64 | 70.95 | 95.54 | 77.66 | 95.11 | 11.2M | 0.7G |
| | 64 | 70.10 | 94.52 | 76.19 | 94.90 | 2.6M | 0.2G |
| Res-18(SPC) | 96 | 71.88 | 95.72 | 78.35 | 95.33 | 5.7M | 0.4G |
| | 128 | **73.24** | **95.84** | **79.77** | **95.54** | 10.2M | 0.8G |

convolution with only half of the parameters (Res-18(SPC), $N_C$=96). Increasing $N_C$ to 128, the Res-18(SPC) reaches similar computational costs to the baseline Res-18 while achieving 4.7% higher accuracy. Similar trends can be observed on small-scale recognition tasks, as shown in Table 11. This confirms that the SPC module can also be used as the main component to construct neural networks, potentially serving as an alternative to convolutional layers in independent manners.

## 5 Conclusion

This paper proposes the SPC module that conducts the Pillars-Shift and Pillars-Concatenation to achieve an elaborate and parallelizable aggregation of local information, with superior classification performance than convolutional layers. Based on SPC, we introduce Caterpillar, a pure-MLP network that attains impressive scores on both small- and large-scale image recognition tasks.

The philosophy of "simple and effective" and the principle of "control variable" have run through this work. Therefore, Caterpillar only replaces the DWConv with SPC module in sMLPNet and thus has more parameters. We anticipate that integrating the SPC module with lightweight techniques, like depth-wise, will further reduce computational costs. Additionally, the experiments are primarily conducted on the most fundamental classification tasks, since SPC and Caterpillar are introduced for the first time. We hope the SPC and Caterpillar can be explored in broader tasks like detection and segmentation, particularly in data-hungry domains.

## 6 Acknowledgments

This work was supported by the National Key Research and Development Program of China under Grant (No. 2022YFA1004100), and the National Natural Science Foundation of China (No. 62276052).

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
