# OpenReview forum: "Caterpillar: A Pure-MLP Architecture with Shifted-Pillars-Concatenation"
_acmmm.org/ACMMM/2024/Conference — MM2024 Poster_

### Official Review · Reviewer_d1sZ · 2024-05-15

**Rating:** 4
**Confidence:** 4

**Summary:**

This paper proposes a novel MLP-based module called Shifted-Pillars-Concatenation (SPC) for local modeling in computer vision. The SPC module consists of two key processes: 1) Pillars-Shift, which generates four neighboring feature maps by shifting the input image along four directions, and 2) Pillars-Concatenation, which applies linear transformations and concatenation on the maps to aggregate local features. Building upon the SPC module, the authors construct a pure-MLP architecture named Caterpillar by replacing the convolutional layers. Experiments demonstrate that Caterpillar achieves excellent performance on both small-scale datasets without extra data and the large-scale ImageNet-1k benchmark. The SPC module also exhibits strong local modeling capability and brings significant performance gains when replacing convolutional layers.

**Strengths:**

**(S1) The Research Question:** The paper addresses an important problem in vision MLPs: exploiting local information in a more elaborate and parallelizable way. The SPC module captures local features from neighboring pillars in parallel, preventing the limitations of convolution, such as redundancy and sequential computation. This is a key contribution that addresses limitations of existing convolution models and MLPs.

**(S2) The Experiments:** Extensive experiments on both small-scale datasets and the large-scale ImageNet-1k benchmark comprehensively validate the effectiveness of Caterpillar and the SPC module. The Caterpillar architecture, built upon the SPC module, consistently achieves superior or competitive performance compared to several existing MLP, ConvNets, and Transformer models. Moreover, the significant performance gains obtained by replacing convolution with SPC in existing architectures further demonstrate the plug-and-play potential of SPC module. The paper also conducts detailed ablation studies to justify the architectural design choices in SPC and Caterpillar, further evaluating the impact of key hyper-parameters such as shift directions, shift steps, padding modes, and local-global combination strategies. These ablations provide valuable insights into the working mechanism and the robustness of this work.

**(S3) The Presentation Clarity:** The paper is overall well-structured and follows a clear logical flow. Necessary background information on vision MLPs is provided, setting the context for their contributions. The methodology section explains the technical details of the SPC module and Caterpillar architecture in a coherent manner, accompanied by illustrative figures and algorithmic descriptions. The extensive experimental results are thoroughly reported and analyzed, with informative tables and plots to support the findings. Visualizations of feature maps learned by SPC and convolution are also presented, facilitating a deeper understanding of their differences in local feature learning. Overall, the clear writing style, detailed methodology, and comprehensive experiments enhance the readability and reproducibility of the paper.

**Limitations:**

### **Limitations:**

**(L1) Efficiency Concerns:** Although the SPC module demonstrates competitive performance, it introduces additional parameters and computational costs. The authors acknowledge the importance of improving the efficiency of SPC for practical usage and discuss potential solutions, such as integrating depth-wise convolution techniques. However, the paper lacks concrete implementations or evaluations of these efficiency optimizations. Providing empirical results or theoretical analysis on the computational complexity, such as total training & inference time, latency, throughput, and memory footprint of SPC would strengthen the practical value of this work. Exploring and demonstrating effective strategies to reduce the overhead of SPC while maintaining its performance gains is crucial for its wider adoption in resource-constrained scenarios.

**(L2) Insufficient Discussion and Comparisons:** While the paper compares Caterpillar with several existing models, it misses comparisons with some recently published state-of-the-art convolutional models. For example, on the ImageNet-1K benchmark, including models like VAN [1], SLaK [2], and MogaNet [3] would provide a more comprehensive evaluation of Caterpillar's performance against modern ConvNets. These models have demonstrated strong results and introduce novel architectural designs or training strategies. Thus, comparing Caterpillar with these models and discussing their similarities and differences would help position the proposed approach in the broader context of vision architecture research. Additionally, the paper would benefit from an expanded discussion section that analyzes the recent developments and trends in ConvNets, highlighting the advantages and limitations of the proposed MLP-based approach.

**(L3) Limited Experiment Scope:** The experimental evaluation in the paper primarily focuses on image classification tasks. While classification is a fundamental problem in computer vision, the versatility and potential of Caterpillar and SPC could be further demonstrated by applying them to a broader range of vision tasks, such as object detection and semantic segmentation. The authors mention this as a future direction, but including initial experiments or a more detailed discussion on the potential of Caterpillar and SPC in these domains would enhance the impact and significance of the work.

### Reference

[1] Visual Attention Network, CVMJ 2023

[2] More ConvNets in the 2020s: Scaling up Kernels Beyond 51x51 using Sparsity, ICLR 2023

[3] MogaNet: Multi-order Gated Aggregation Network, ICLR 2024

---

### **Justification of Rating:**

After carefully reading the manuscript, I first recommend a Borderline Accept for this paper.

**(1) Research Questions**

This paper tackles the important problem of local feature learning in vision MLPs. It identifies the limitations of convolution in terms of redundancy and sequential computation, and proposes the SPC module as a more effective and parallelizable alternative. The research questions are well-motivated and clearly stated.

**(2) Challenge Analysis**

The authors provide a thorough analysis of the challenges in local modeling with vision MLPs. They discuss the drawbacks of existing solutions such as traditional convolution and previous MLP-based approaches. This analysis justifies the necessity of the proposed SPC module and Caterpillar architecture.

**(3) Philosophy**

The key philosophy behind this work is to exploit locality in vision MLPs in a more elaborate and efficient way. The SPC module achieves this by encoding local features from neighboring pillars in parallel, avoiding the sliding window scheme. This philosophy aligns well with the goal of designing effective vision MLPs.

**(4) Technical Soundness and Originality**

The proposed SPC module and Caterpillar architecture are technically sound and original. SPC introduces a novel way to aggregate local features through Pillars-Shift and Pillars-Concatenation operations, demonstrating superior performance over convolution. Caterpillar leverages SPC to build a pure-MLP model that separately captures local and global information, leading to excellent results.

**(5) Evaluation**

The experiments are extensive. The authors evaluate Caterpillar on both small-scale datasets and ImageNet-1k, comparing it with several MLP, CNN, and Transformer models. However, it is better to discuss and empirically compare the missing recent convolutional methods as I stated above. Ablation studies are conducted to justify the design choices. The consistent superior performance of Caterpillar and the significant gains brought by replacing convolution with SPC  support the effectiveness of the proposed methods.

**(6) Presentation Clarity**

The paper is well-structured and clearly written. The motivation, methodology, and experiments are presented in a logical flow. Necessary details are provided for reproducibility. The illustrations and visualizations help understand the technical contributions.

Considering the above factors, I believe the paper in its current form meets the bar for a Borderline Accept at ACM MM 2024.

---

### **Additional Comments:**

I hope my review helps to further strengthen this paper and helps the authors, fellow reviewers, and Area Chairs understand the basis for my recommendation. I look forward to the rebuttal feedback and would be glad to raise my rating if thoughtful responses and improvements are presented.

**Suitability:**

2

---

### Official Review · Reviewer_U832 · 2024-05-15

**Rating:** 5
**Confidence:** 4

**Summary:**

This paper introduces a new MLP-based architecture for computer vision tasks that aims to replace traditional convolutional methods with a pure multi-layer perceptron (MLP) approach. The core innovation, the Shifted-Pillars-Concatenation (SPC) module, enhances the local modeling capability by shifting the input image in four directions and concatenating the results after applying linear transformations. This architecture is tested on a variety of datasets, including ImageNet-1k, showing competitive or superior performance compared to existing models.

**Strengths:**

-   Introducing a pure-MLP architecture that does away with convolutions and instead uses pillar shifting and concatenation is a significant deviation from traditional models, offering a fresh perspective in the field.

-   The proposed module has the ability to capture local dependencies without convolution providing a promising alternative for reducing computational redundancy.

-   Experiments show the effectiveness of the proposed method and the performance is promising.

**Limitations:**

The authors propose to use Pillars-Shift to achieve the local receptive fields, which is a good idea. Some other papers such as AS-MLP [25] also utilize a similar operation to shift features along two directions (horizontal and vertical) to cover the local receptive fields. The author could highlight the main differences between the proposed method and AS-MLP. In addition, as we know, although the parameters and flops are the metrics to measure the complexity of a model, the inference time (or throughput) is a more direct way. The authors could also list the throughput of different models in Table 1 or Table 2 to make the experimental results more complete. As pointed out in the conclusion, some experiments on detection or segmentation tasks are necessary because the classification task only needs a small receptive field to recognize an image but the detection or segmentation tasks need a larger receptive field dependency due to the property of the tasks and larger resolution images.

This work proposes the SPC module that conducts the Pillars-Shift and Pillars-Concatenation to achieve an elaborate and parallelizable aggregation of local information. Experimental results on the classification datasets demonstrate the effectiveness and efficiency of the proposed approach. My questions are listed above. I tend to vote weak accept.

**Suitability:**

2

---

### Official Review · Reviewer_JgkS · 2024-05-20

**Rating:** 2
**Confidence:** 4

**Summary:**

This paper propose a MLP module which exploit local modeling capability. The main contributions of the paper include proposing a new SPC module and constructing a new pure-MLP model Caterpillar, which is able to perform superior or better on image datasets without relying on Transfer Learning.

**Strengths:**

-This paper introduces local modeling capabilities for MLPs. The SPC module proposed in this paper adopts a windowless scheme and can exploit locality in a more refined and parallel way than convolution.

-The writing of this paper is well, and the organization of the paper is good.

**Limitations:**

-The paper does not discuss in depth enough the comparison of the parameter efficiency and computational complexity of the SPC module and the traditional convolutional.

-The improvement in model performance is not significant, and many additional parameters are introduced. MLPs have no advantage in terms of computational complexity.

-The authors could provide theoretical analysis for the proposed method and comparison algorithms.

-The empirical comparison can be done in a more thoroughly by comparing with other latest state-of-the-art algorithms.

**Suitability:**

2

---

### Official Review · Reviewer_NkQz · 2024-06-06

**Rating:** 2
**Confidence:** 4

**Summary:**

The paper introduces the Caterpillar architecture, a pure-MLSLP structure enhanced by the Shifted-Pillars-Concatenation (SPC) module, designed to improve local feature integration without reliance on traditional convolutional methods. This model leverages shifts and linear transformations across image pillars to efficiently capture and concatenate local spatial information, demonstrating robust performance on various image datasets.

**Strengths:**

1. Caterpillar achieves impressive results on benchmarks like ImageNet-1k, showcasing its potential against established MLP, CNN, and Transformer models.
2. The architecture is designed to be computationally efficient, reducing the parameter count and computational overhead compared to conventional methods.
3. The paper is well-written and easy to follow.

**Limitations:**

1.The shift operation is not novel in the vision community, such as [1] and [2]. More detailed comparison and discussion with these methods are needed. The novelty also needs further clarification.

[1]. A Simple Baseline for Video Restoration with Grouped Spatial-temporal Shift
[2]. X-volution: On the Unification of Convolution and Self-Attention

2.Comparison with SOTA methods is missing.
[3] A ConvNet for the 2020s
[4] MLP-Mixer: An All-MLP Architecture for Vision
[5] InternImage: Exploring Large-Scale Vision Foundation Models with Deformable Convolutions
[6] HorNet: Efficient High-Order Spatial Interactions with Recursive Gated Convolutions.
[7] CYCLEMLP: A MLP-like Architecture for Dense Prediction

3.As a foundation model, experiments on detection and segmentation are missing, which makes the validity of the method less credible.

4. The effective receptive field of the method needs to be analyzed. Also, the feature map visualization is insufficient.

5. Why the shift operation works needs to be further explained.

**Suitability:**

3

---

### Meta-Review · Area_Chair_SCxQ · 2024-07-02

**Recommendation:** Accept (Poster)
**Confidence:** 4

**Metareview:**

This paper proposes Caterpillar, a pure-MLP architecture with a Shifted-Pillars-Concatenation (SPC) module aimed at enhancing local feature modeling in computer vision tasks.

Initial concerns mainly lie on the novelty of the shift operation, the lack of comprehensive comparison with state-of-the-art methods, insufficient discussion on parameter efficiency and computational complexity, and the limited scope of experiments focusing primarily on image classification tasks.

The authors provided a thorough response addressing the concerns raised by the reviewers, leading to a final scores of two Weak Reject and two Weak Accept, with an average score of 3.5.

The AC carefully reviewed the paper and all the reviewers' comments.
Considering the good performance of the proposed architecture on benchmark datasets and its potential impact for the community, the AC agrees to accept the paper and strongly recommends incorporating the content from the rebuttal into the final version.